# Hypervector Design for Efficient Hyperdimensional Computing on Edge Devices

## ABSTRACT

Hyperdimensional computing (HDC) has emerged as a new light-weight learning algorithm with smaller computation and energy requirements compared to conventional techniques. In HDC, data points are represented by high dimensional vectors (hypervectors), which are mapped to high dimensional space (hyperspace). Typically, a large hypervector dimension ($\geq 1000$) is required to achieve accuracies comparable to conventional alternatives. However, unnecessarily large hypervectors increase hardware and energy costs, which can undermine their benefits. This paper presents a technique to minimize the hypervector dimension while maintaining the accuracy and improving the robustness of the classifier. To this end, we formulate hypervector design as a multi-objective optimization problem for the first time in the literature. The proposed approach decreases the hypervector dimension by more than $128\times$ while maintaining or increasing the accuracy achieved by conventional HDC. Experiments on a commercial hardware platform show that the proposed approach achieves more than two orders of magnitude reduction in model size, inference time, and energy consumption. We also demonstrate the trade-off between accuracy and robustness to noise and provide Pareto front solutions as a design parameter in our hypervector design.

## CCS CONCEPTS

• **Computing methodologies** → **Machine learning**; • **Hardware** → *Power and energy*.

## KEYWORDS

Hyperdimensional computing, low-power learning algorithm, optimization

## 1 INTRODUCTION

The number of Internet of Things (IoT) devices and the data generated by them increase every year [1, 2]. Wearable IoT, which utilizes edge devices and applications for remote health monitoring [3], has been a rapidly growing subfield of IoT in recent years [4]. These devices fuse data from multiple sensors, such as inertial measurement units (IMU) and biopotential amplifiers, to achieve accurate real-time tracking. For example, assistive devices for Parkinson's Disease patients need to provide precisely timed audio cues to cope with gait disturbances [5]. Despite the intensity of sensor data,

*tinyML '21, March 22–26, 2021, Online*
© 2020 Association for Computing Machinery.
ACM ISBN 978-1-4503-XXXX-X/21/03...$15.00
https://doi.org/10.1145/nnnnnnn.nnnnnnn

these devices must operate at a tight energy budget ($\sim \mu$W) due to limited battery capacity [6]. Offloading the data to the cloud is not an attractive solution since it increases the communication energy [7] and raises privacy concerns [8]. Likewise, deep neural networks and other sophisticated algorithms are not viable due to the *limited computation and energy capacity of wearable edge devices*. Therefore, there is a strong need for *computationally light* learning algorithms that can provide high accuracy, real-time inference, and robustness to noisy sensor data.

Brain-inspired hyperdimensional computing [9] provides competitive accuracy to state-of-the-art machine learning (ML) algorithms with significantly lower computational requirements for various applications, such as human activity recognition, language processing, image recognition, and speech recognition [10]. It models the human short-term memory [9, 11] using high dimensional representations of the data points, which is motivated by the large size of neuronal interactions in the brain to *associate* a sensory input with the human memory. *The main difference between HDC and conventional learning approaches is the primary data type.* HDC maps data points (raw samples or extracted input features) in the input space to random high-dimensional vectors, called *sample hypervectors*. Then, the sample hypervectors that belong to the same class are combined linearly to obtain ensemble class hypervectors, called *class encoders*. During inference, the input data is used to generate a *query hypervector* ($Q$) in the same way as the sample hypervectors. The classifier simply finds the closest class hypervector to $Q$, generally using cosine similarity or the Hamming distance.

Thanks to the simplicity of binary operations, as discussed in Section 3, HDC lends itself to efficient hardware implementation. A recent study [12] shows that custom hardware implementations can provide high energy efficiency and inference speed while reducing the design complexity. However, HDC requires a large dimension (e.g. $D \geq 1000$) to achieve a high inference accuracy [13] due to the *random mapping* process. A sufficiently large dimension is needed to ensure, with high probability, that the sample hypervectors are orthogonal to each other [10, 12]. A larger dimension implies higher energy consumption, longer inference time, and more hardware resources [13]. Consequently, redundant computations can undermine the benefits of using HDC over other learning algorithms. Therefore, there is a critical need to *optimize the design of hypervectors* such that the performance of HDC is maintained with smaller dimensions.

This paper presents a novel optimization algorithm for representing the input data points in the hyperspace, *instead of relying on random mapping*. We conceptualize the mapping of the data points in the hyperspace by geometric notions. Using this insight and a novel *non-uniform quantization* approach, we refine the distribution of randomly generated sample hypervectors in the hyperspace to achieve increased robustness to noise in smaller dimensions and similar accuracy levels to that of conventional HDC. To the

best of our knowledge, this is the first technique that formulates *hypervector design* as a multi-objective optimization problem to achieve higher accuracy and robustness using smaller dimensions (i.e., lower energy and computational resources). The proposed approach is evaluated using four representative health-oriented applications since health monitoring is an emerging and attractive field in wearable IoT literature: Parkinson's Disease diagnosis, electroencephalography (EEG) classification, human activity recognition, and fetal state diagnosis.

*The major contributions of the proposed approach are as follows*:

- It boosts the effectiveness of HDC by enabling more than two orders of magnitude model compression while maintaining similar performance to conventional HDC. Furthermore, hardware measurements show over 600× higher energy efficiency.
- It introduces novel geometric concepts and illustrations to better understand the effect of HDC mapping from the input features to the hyperspace.
- By optimizing the trade-off between accuracy and robustness to noise, it achieves 2× higher robustness while maintaining the same accuracy, or 1%–18% higher accuracy without sacrificing the robustness.

In the rest, Section 2 reviews the related work, while Section 3 overviews HDC. Section 4 presents the proposed hypervector design optimization. Section 5 presents the evaluation of the proposed approach and provides results on four applications. Finally, Section 6 concludes the paper.

## 2 RELATED WORK

HDC utilizes high dimensional vectors to map the input features to high dimensional space. It achieves competitive results compared to other state-of-the-art ML approaches [10]. However, HDC requires large dimensions ($D \geq 1000$) to achieve high inference accuracy due to the random mapping process. In turn, large dimensions increase on-chip storage, computation, and energy requirements, which are limited in wearable edge devices. Hence, there is a critical need to optimize the mapping process in HDC to achieve similar or higher accuracy and robustness with smaller dimensions.

One idea to remedy high dimensional data is using dimensionality reduction techniques, which are linear/nonlinear transformations that map the high-dimensional data to a lower-dimensional space. Recently, nonlinear dimensionality reduction techniques or manifold learning algorithms, such as ISOMAP, local linear embedding (LLE), and Laplacian eigenmaps, have become widely used for dimensionality reduction [14]. These algorithms mainly focus on *preserving the geometric information* of the high-dimensional data in the low dimensional space in contrast to another widely used linear technique, principal component analysis (PCA), which may distort the local proximities by mapping the data points that are distant in the original space to nearby positions in the lower dimensional space [15]. However, the main drawback of the manifold learning methods is learning the low-dimensional representations of the high-dimensional input data samples implicitly. Explicit mapping from the input data manifold to the output embedding cannot be obtained after the training process [14, 16] without compromising the memory and computational limitations of a wearable edge device. For instance, the ARM Cortex-M series is a widely used family of low-power processors for wearable edge devices [17]. This family offers an on-chip SRAM of a few KB and nonvolatile flash memory with a size of up to 2 MB [18]. Considering a large amount of training data and the dimensionality in HDC ($D \geq 1000$), the memory footprint of these low-power processors is not sufficient to obtain an explicit mapping. Therefore, an optimized representation of the data in smaller dimensions is necessary for these devices.

A recent work investigated the impact of dimensionality on the classification accuracy and energy efficiency of HDC [13]. The authors show that energy consumption and inference time decreases with smaller dimensions. However, smaller dimensions yield lower accuracy values. Hersche *et al.* propose a mapping technique based on the training of random projection to produce distinct hypervectors as part of learned projections [19]. The learned projections are more effective in terms of accuracy at lower dimensions. However, this mapping does not preserve the level dependency between level hypervectors in HDC, which is critical for robust classification, as elaborated in Section 4.1. Another study reduces the high dimensionality of the class encoders by dividing it into multiple smaller dimensional segments and adding them up to form a lower-dimensional class encoder [20]. However, during inference, the query hypervector is formed using high dimensional level hypervectors that need to be stored in memory and thus undermine the potential energy savings.

In contrast to prior work, this paper focuses on optimizing hypervector design at the initial stage of HDC training. We formulate the hypervector design as a multi-objective optimization problem for the first time in literature. Moreover, none of the prior studies discuss the robustness to the noise of the HDC model. The proposed approach produces an efficient and robust representation of the data points in the hyperspace while preserving similarity between level hypervectors. It also is the first work that enables a trade-off between robustness and accuracy by providing a Pareto front set of hypervector design.

## 3 BACKGROUND ON HDC

### 3.1 HDC Training

Training in HDC consists of three steps: ① Quantization and mapping , ② Construction of sample hypervectors and ③ Classifier encoding, as illustrated in Figure 1. We refer to this architecture as the *baseline HDC implementation* in the rest of the paper.

① **Quantization and mapping:** The first step of HDC is to quantize the input space and represent it using **D**-dimensional *level hypervectors*. This is achieved in two steps: (i) quantization in low dimension, and (ii) mapping to high dimension.

**Quantization in low dimension:** Let $F = \{f_1, f_2, ..., f_N\}$ denote the $N$-dimensional input feature, where $f_n \in \mathbb{R}$ corresponds to the $n^{th}$ feature. Suppose that there are $S$ training samples. Each training sample $x_s$ for $s \in \{1, ..., S\}$ is an $N$-tuple $x_s = \{x_s^1, x_s^2, ..., x_s^N\}$, where $x_s^n \in \mathbb{R}$ is the value of feature $f_n$ in this sample for $n \in \{1, ..., N\}$.

Using the training set, we find the minimum ($f_n^{\min}$) and the maximum ($f_n^{\max}$) values of each feature. Let $Q = \{1, ..., M\}$ be the set of quantization levels. The baseline HDC quantizes each feature space into $M$ uniform levels using a quantization function $q : \mathbb{R} \rightarrow Q$ such that $q(x_s^n)$ returns the quantization level for feature $f_n$ in input sample $x_s$.

**Table 1: Notation table. \*HV: Hypervector**

| Symb. | Description | Symb. | Description | Symb. | Description | Symb. | Description |
|---|---|---|---|---|---|---|---|
| $N$ | # of features | $q(.)$ | Quantizer function | $x_s$ | Training sample $s$ | $L_n^m$ | $D$-dim level HV* for $m^{th}$ level of $f_n$ |
| $F$ | $N$-dim input space | $S$ | # of training samples | $X_s$ | $D$-dim sample HV* for $x_s$ | | |
| $f_n$ | $n^{th}$ dimension of $F$ | $D$ | Hyperspace dimension | $Q$ | $D$-dim query HV* | $b$ | # of flipped bits between $L_n^m$ and $L_n^{m+1}$ |
| $M$ | # of quantization levels | $g(.)$ | Nonlinear bit-flip function | $K$ | # of different classes | | |
| $y_s$ | Class label for $x_s$ | $E_k$ | Encoder for $k^{th}$ class | $Q$ | Set of quantization intervals | | |

**Mapping to the high dimension:** The minimum level of each feature $f_n^{min}$ is assigned a random bipolar $D$-dimensional hypervector $L_n^1$, where $L \in \{-1, 1\}^D$, and typically $D \geq 1000$. The rest of the level hypervectors $L_n^2$ to $L_n^M$ are calculated by randomly flipping $b = D/2(M-1)$ bits to generate each consecutive hypervector starting from $L_n^1$. This operation is denoted by $L_n^{m+1} = g(L_n^m, b) \; \forall \, m \in Q$, where $g(u, v) : \{(u, v) | \, u \in \{-1, 1\}^D, v \in \mathbb{N}\} \rightarrow \{-1, 1\}^D$ is a nonlinear function that flips $v$ (scalar) indices of $u$ (vector). This process tracks the flipped bits and ensures that they do not flip again in subsequent levels. At the end, it produces $N \times M$ different *level hypervectors* $L_n^m \; \forall \, n, m$ that represent the quantized features in the hyperspace. This process also ensures that hypervectors $L_n^1$ and $L_n^M$ orthogonal for each feature $f_n$ [10]. The notation is summarized in Table 1.

② **Construction of sample hypervectors ($X_s$):** The next step is constructing the sample hypervectors using the input samples $x_s$ and level hypervectors found in step ①. We first determine the quantization levels $q(x_s^n)$ that contain the value for each feature of the current sample $x_s$. Then, we fetch the level hypervectors $L_n^{q(x_s^n)}$ that correspond to these quantization levels and add them up:

$$X_s = \sum_{n=1}^{N} L_n^{q(x_s^n)} \; \forall \, s \in \{1, 2, ..., S\}, \text{ where } X_s \in \mathbb{Z}^D \tag{1}$$

**Toy example:** For a better understanding of steps ① and ②, we present a simple representative example. Consider a 2-D problem with $f_1 \in [0, 1]$ and $f_2 \in [-10, 0]$ with $M = 10$ quantization levels and $D = 1000$. The quantization intervals between $f_1^{min} = 0$ and $f_1^{max} = 1$ are $[0, 0.1), [0.1, 0.2), ..., [0.9, 1]$. Similarly, the quantization intervals for $f_2$ are $[-10, -9), [-9, -8), ..., [-1, 0]$. The intervals $[0, 0.1)$ and $[-10, -9)$ are assigned random bipolar 1000-dimensional hypervectors $L_1^1$ and $L_2^1$. To find the rest of the level hypervectors $L_1^2$ to $L_1^{10}$ and $L_2^2$ to $L_2^{10}$, we first calculate the number of bits to flip at each consecutive level as $b = D/2(M - 1) = 55$.

Then, $L_1^2$ is found by $L_1^2 = g(L_1^1, 55)$ and the rest of $L_1^m$ and $L_2^m$ are calculated through the same procedure. As a result, we obtain 20 different *level hypervectors* that represent the quantized $f_1$ and $f_2$ in the hyperspace.

After finding the level hypervectors, input samples $x_s$ are mapped to their corresponding hypervectors $X_s$. For example, consider an arbitrary sample $x_3 = \{0.17, -1.2\}$. We first find the quantization levels 0.17 and −1.2 fall into. These levels are given by $q(x_3^1) = 2$ and $q(x_3^2) = 8$. Then, we find $X_3 = L_1^2 + L_2^8$ using Equation 1. This procedure is repeated for all $x_s$ in the training set.

③ **Classifier encoding:** Suppose that there are $K$ classes with labels $1, 2, ..., K$. The label of sample hypervector $X_s$ is given by $y_s \in \{1, ..., K\}$ for all samples $s \in \{1, ..., S\}$. The class hypervector $E_k \in \mathbb{Z}^D$ that represents class $k$ is found by adding all the sample hypervectors with label $k$:

$$E_k = \sum_{s=1}^{S} X_s [y_s = k] \tag{2}$$

where $[.]$ operation represents the Iverson Bracket that is equivalent to the indicator function [21].

## 3.2 HDC Inference

During inference, the query samples with unknown class labels are first mapped to the hyperspace using the procedure defined in steps ① and ②. The resulting hypervectors are called *query hypervectors* $Q$. Then, the cosine similarity between the query hypervector and each class encoder $E_k$ given in Equation 2 is calculated. Finally, the class $k$ with the highest similarity is decided as the class label of the query data point as follows:

$$\underset{k \in \{1, ..., K\}}{\arg \max} \frac{Q \cdot E_k}{\|Q\| \, \|E_k\|} \tag{3}$$

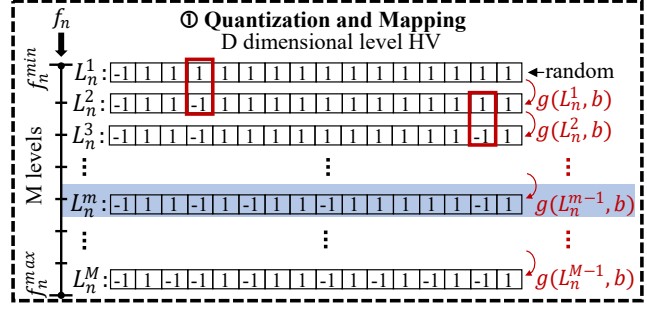
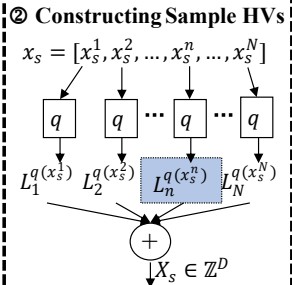
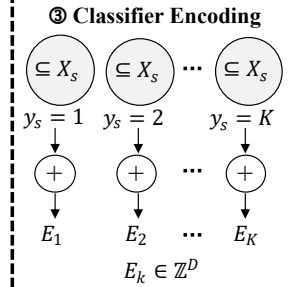

**Figure 1: Overview of baseline HDC. $D = 16$ and $M = 8$ are chosen for illustration purposes. Bit values are chosen arbitrarily.**

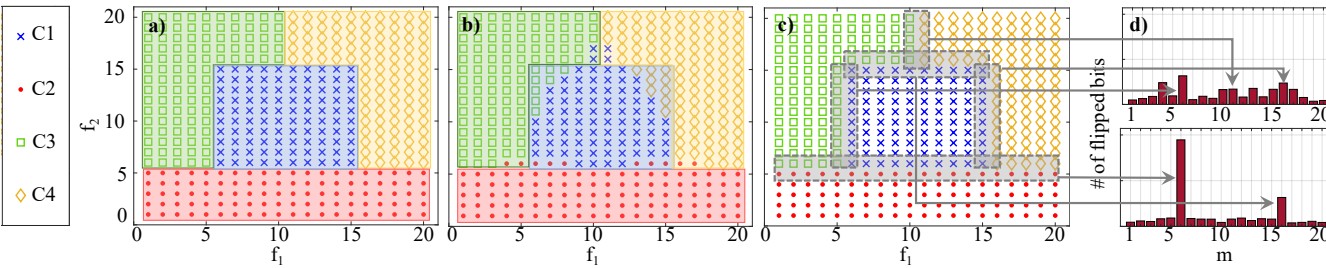

**Figure 2: Evaluation of the proposed approach on a motivational example. a) Initial problem structure. b) Baseline HDC ($D$ = 8192) classification result. c) HDC classification result obtained by separating significant levels for both features ($D$ = 64). d) Number of bits to flip for each feature and level.**

## 4 OPTIMIZED HYPERVECTOR DESIGN

### 4.1 Motivation

**Distance in the hyperspace:** Two samples from different classes may be close to each other, in the low dimensional input space, based on the extracted features or raw data. This proximity eventually decreases the accuracy and robustness of the classifier. Using a geometrical interpretation, we can consider the sample hypervectors, *i.e.,* the mapping to the high dimensional domain, as data points in the hyperspace. Using this insight, the optimized hypervector design should achieve two objectives: 1) *Spread sample hypervectors from different classes as far as possible*, 2) *Cluster sample hypervectors of the same class close to each other*.

**Dependency between feature levels:** One can assign orthogonal hypervectors to each level hypervector during quantization to spread them far apart [22]. This approach can provide high classification efficiency *if all levels are represented in the training data*. However, if the training data has gaps, *e.g.*, certain levels are underrepresented, then the classifier can make random choices. Hence, the hypervector optimization technique must maintain the dependency between the level hypervectors $L_n^m$ for each feature $n \in \{1, 2, ..., N\}$. We provide a mathematical illustration with a simple example in Appendix to not distort the flow of the paper.

### 4.2 Illustration using a Motivational Example

This section presents a motivational example to illustrate the significance of the proposed hypervector optimization. In this example, the data points are divided into four classes (C1, C2, C3 and C4) and represented by two features $F = \{f_1, f_2\}$, as shown in Figure 2(a).

For the baseline HDC implementation, we choose $D$ = 8192 and $M$ = 20 and follow the procedure explained in Section 3 to represent the data points in the hyperspace. All data points are included in the training and test set for illustration purposes. The baseline HDC misclassifies a significant number of data points concentrated around the class boundaries, as shown in Figure 2(b). The specification in Figure 2(a) shows that the levels 6, 11, and 16 are critical for the first feature ($x$−axis), while level 6 and 16 are critical for the second feature ($y$−axis). However, the baseline HDC ignores this fact and quantizes the levels uniformly, leading to misclassified points at the boundaries.

In contrast to uniform quantization levels, the proposed approach emphasizes the distinctive levels in the training data. For example, Figure 2(d) shows that the number of bits flipped by the proposed

approach between consecutive levels. The lower plot clearly shows that it allocates a significantly higher number of bits for levels 6 and 16, which precisely matches with our earlier observation. In general, the proposed approach flips a different number of bits at each consecutive level as opposed to the uniform $b = D/2(M − 1)$ bits used by the baseline HDC. As a result, it achieves 100% classification accuracy by judiciously separating the quantization levels of both dimensions, as illustrated in Figure 2(c). More remarkably, the proposed approach achieves a higher level of accuracy than the baseline HDC by using only $D$ = 64 dimensions, as opposed to 8192. This result indicates that the number of dimensions can be compressed significantly while increasing the accuracy through optimized hypervector design.

We also demonstrate a geometrical illustration on the same example using 2-D t-distributed stochastic neighbor embedding (t-SNE) [23] representation of the sample hypervectors. t-SNE is a nonlinear dimensionality reduction algorithm that works especially well in visualizing high-dimensional data points. Figure 3(a) shows that the baseline HDC performs poorly in the separation of hypervectors for different classes despite the large number of dimensions ($D$ = 8192). In contrast, Figure 3(b) shows that the optimized sample hypervectors are separated well from each other according to their classes while using a much smaller $D$ = 64. The t-SNE and accuracy results illustrate the importance of the objectives set for the hypervector optimization presented in the next section.

### 4.3 Optimization Problem Formulation

This section formulates the level hypervector design as an optimization problem where the number of bits to flip between each consecutive level hypervector are the optimization parameters. In

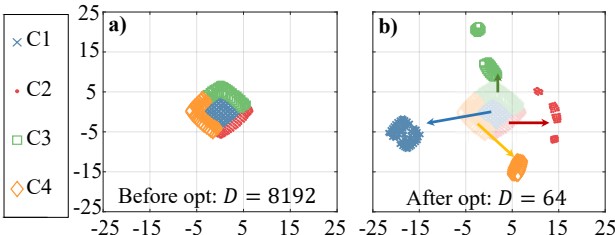

**Figure 3: t-SNE illustration of sample hypervectors in 2-D for the motivational example.**

contrast to baseline HDC which uses a uniform number of levels as $b = D/2(M-1)$, we use a variable number of bits $b_n^m \ \forall \ n \in \{1, ..., N\}$ and $m \in \{1, ..., M\}$. The optimization parameters are defined as a matrix $B_{N \times M}$, which includes all $b_n^m$ values for $N$ features and $M$ levels. Suppose that there are $K$ classes in the dataset, and $TP_k$ and $FN_k$ represent the true positives and false negatives for class $k$, respectively. We construct the following multi-objective optimization problem with two objectives: (i) maximize the training accuracy and (ii) minimize the similarity between class encoders:

$$\max_{B_{N \times M}} \ wAcc = \frac{1}{K} \sum_{k=1}^{K} \frac{TP_k}{TP_k + FN_k} \tag{4}$$

$$\min_{B_{N \times M}} \ avgSim = \left( \prod_{k=1}^{K} \prod_{k'=1}^{K} \frac{E_k \cdot E_{k'}}{\|E_k\| \, \|E_{k'}\|} \right)^{\frac{1}{K}} \ \text{for} \ k \neq k' \tag{5}$$

$$\text{subject to} \quad \sum_{m=1}^{M} b_n^m \leq \frac{D}{2} \quad \text{where} \ n \in \{1, ...N\} \tag{6}$$

where Equation 6 ensures that the total number of bits flipped between $L_n^1$ and $L_n^M$ does not exceed $D/2$. This condition is critical to satisfy the orthogonality condition between distant values in the input space, as explained in Section 3.1. The proposed formulation uses the weighted accuracy ($wAcc$) instead of the total accuracy since the total accuracy calculation suffers from imbalanced datasets. Equation 5 formulates the geometric mean of cosine similarity between each class encoder pairs. It provides a lumped similarity metric between class encoders. The effect of outliers (e.g. one class encoder is significantly different from the others) is greatly dampened in the geometric mean. The distance between different class encoders (i.e., $1 - avgSim$) is used as a measure of robustness and maximized by Equation 5.

## 4.4 Optimization Problem Solution

The objective functions in the proposed formulation (Equations 4 and 5) are nonlinear functions that can be evaluated only at integral points. Furthermore, there are integer constraints since the optimization parameters are the number of bits flipped between each consecutive level hypervector. Hence, we need to solve a non-convex optimization problem with integer constraints, i.e., an NP-hard problem. One can employ gradient-based approaches to solve this problem by relaxing the integer constraints (e.g., using continuous variables and rounding them to the nearest integer to evaluate the objective functions). However, gradient-based approaches get stuck at a local minimum near the starting point. This obstruction occurs since the objective functions typically have many minimums, and rounding is used during evaluation. To overcome this limitation, we employed gradient approaches using multiple starting points. Nevertheless, the solutions are still not far from the original starting points, which are not close to the optimal solution. Hence, we conclude that gradient-based approaches are not suitable for our problem. We employ a genetic algorithm (GA), an evolutionary heuristic search approach, to find a solution close to the global minimum. Our aim is not only to obtain the best solution that gives the highest accuracy but also to optimize the trade-off between the accuracy and the robustness of the model. Since GA maintains a population of possible solutions at every generation and is a highly

explorative algorithm, it is preferred in this work, considering the objective functions have many local minima.

Algorithm 1 describes the search approach to find the optimum hypervector design. The input to the algorithm is $P$ randomly generated $B_{N \times M}$ matrices ,where $P$ is the population size in GA. This input corresponds to the first generation in GA. First, the algorithm updates the level hypervectors based on the $b_n^m \ \forall \ n, m$ for each population. Then, the sample hypervectors and class encoders are updated using the new level hypervectors. Next, the objective functions are evaluated based on the updated hypervectors. Next, GA selects a new set of populations used in the next generation according to the values of the objectives. After a predefined number of generations, we obtain the Pareto front $B_{N \times M}$ set as an output of the algorithm. Our proposed approach utilizes the *gamultiobj* function of MATLAB (The MathWorks, Inc., Natick, MA, USA). For our evaluations, we use a population size of $P = 1000$ and allow the search to run for at least 200 generations with a fixed seed.

---

**Algorithm 1:** Optimized hypervector design with GA

**Input:** Randomly generated $B_{N \times M}$ matrices for each population
**Output:** Pareto front $B_{N \times M}$ set
$G$ = number of generations in GA
$\mathcal{P}$ = set of all populations
$\mathcal{L}$ = set of all level hypervectors
$\mathcal{X}$ = set of sample hypervectors
$E$ = set of class encoders
**for** $i = 1: |G|$ **do**
    **for** $j = 1: |\mathcal{P}|$ **do**
        $B_{N \times M} \rightarrow \mathcal{L}, \mathcal{L} \rightarrow \mathcal{X}, \mathcal{X} \rightarrow E$
        Compute $wAcc$ and $avgSim$
        Compute feasibility of the current solution
    Obtain new $\mathcal{P}$ for the next generation
Obtain Pareto front $B_{N \times M}$ set

---

# 5 EXPERIMENTS

## 5.1 Benchmark Applications

We perform our evaluations on four publicly available representative wearable health applications: Parkinson's Disease digital biomarker DREAM Challenge [24], EEG error-related potentials [25], human activity recognition [26], and cardiotocography [27].

- **Parkinson's Disease digital biomarker DREAM challenge dataset (PD Challenge)** provides the mPower dataset, which includes 35410 walking tasks with expert labels (positive or negative diagnosis). For each task, the winning team extracted 57 features that can be used by standard supervised learning algorithms, such as SVM, for accurate classification [24]. These features and the training/test data provided by this challenge are used to evaluate the proposed approach.
- **EEG error-related potentials (ERPs) dataset** is collected from six subjects to study ERPs. User studies consist of two sessions, which are divided into ten blocks. Each block is further divided into 64-2000 ms long trials. During each trial, EEG is recorded from 64 electrodes with a sampling rate of 512 Hz. The first and

second sessions are used for training and testing, respectively. Since the dataset provides raw EEG data, we follow the procedure outlined in [28] for the baseline HDC implementation. The reported accuracy is the average of all six subjects.

- **Human activity recognition (HAR) dataset** provides strech sensor and accelerometer readings of 22 subjects while they are performing eight activities: jump, lie down, sit, stand, walk, stairs up, stairs down, transition. It provides 120 comprehensive features extracted from the raw data. We choose a subset of 33 features using sequential feature selection [29]. The data is divided randomly into 80% training and 20% test set.

- **Cardiotocography (CTG) dataset** provides 21 features from 2126 fetal cardiotocograms, which are extracted to diagnose the fetal state. The fetal state is divided into three classes: normal, suspect, and pathological. We randomly chose 80% of the data as the training set and 20% as the test set.

All the training and test sets used in this work will be released to the public for the reproducibility of our results.

## 5.2 Accuracy – Robustness Evaluation

This section evaluates the proposed approach using the weighted accuracy ($wAcc$) and similarity ($avgSim$) metrics defined in Section 4.3. We first apply the baseline HDC using dimension $D = 8192$, which is shown to be sufficiently large by prior work [13, 20]. Then, the proposed approach is used to minimize the number of dimensions while maintaining the baseline HDC accuracy. For example, baseline HDC achieves 84% weighted accuracy with $D = 8192$ dimensions for the PD Challenge dataset. Our approach reduces the number of dimensions to $D = 32$ (i.e., by 256×) while achieving 86% weighted accuracy, as shown in Figure 4. Similarly, the proposed approach reduces the dimension significantly for other datasets while maintaining or slightly improving the weighted accuracy. Specifically, it decreases the dimension size by 128×, 128×, and 256× for EEG ERP, HAR, and CTG datasets, respectively. Another commonly used metric in multi-class problems is the *total accuracy*, which is the number of correct classifications divided by the total number of samples. Figure 4(c) shows that the proposed approach also successfully maintains the total accuracy.

We also apply a light-weight ML algorithm, linear SVM, to all datasets as a comparison point. HDC achieves competitive accuracy with SVM both for weighted and total accuracy, as shown in Figure 4. Morever, the proposed optimization approach yields

higher weighted accuracy compared to other methods for imbalanced datasets with raw input data. We also note that linear SVM requires additional pre-processing steps on top of the necessary filtering operations, especially for biosignals acquired using many channels/electrodes. For example, canonical correlation analysis is applied to multi-channel EEG data to select the channels with a high signal-to-noise ratio such that the accuracy increases [30]. In contrast, HDC does not require such computationally intensive pre-processing steps.

**Accuracy vs. robustness trade-off:** A unique strength of our approach is optimizing the trade-off between accuracy and robustness, which is not explored by prior work. Robustness is measured by the *dissimilarity* between class encoders defined as $1 - avgSim$ in terms of the average similarity metric introduced in Section 4.3. It is a measure of how distant the clusters for different classes are in the hyperspace. Figure 5 shows the Pareto front solutions in terms of the weighted accuracy ($x$−axis) and dissimilarity ($y$−axis). For all applications, the Pareto front solutions yield a more robust choice of hypervectors than the baseline HDC. For example, the weighted accuracy and robustness of the baseline HDC are (68%, 0.017) with $D = 8192$ dimensions for the EEG ERP dataset, as shown with ∗ in Figure 5(b). The proposed approach increases both weighted accuracy and robustness to (86%, 0.035) using only $D = 64$ dimensions, as shown by the arrow. It can also trade the accuracy off to improve the robustness to as high as 0.08 while still achieving higher accuracy than the baseline HDC. Weighted accuracy and robustness for the Pareto front solutions range from (86%, 0.035) to (78%, 0.08). Similarly, the proposed approach improves weighted accuracy and robustness for other datasets. It improves the weighted accuracy by 3%, 1%, and 6% while also increasing the robustness by 3×, 2×, and 2× for PD Challenge, HAR, and CTG datasets, respectively. At the same time, it can achieve 2×, 2.5×, and 6× higher robustness while maintaining the accuracy of the baseline HDC. In summary, our approach provides a set of hypervector designs to choose from that can be used to achieve a specific objective based on the nature of the application.

## 5.3 Evaluation on Hardware

We evaluate the proposed approach by running the inference for all applications on the Odroid XU3 platform. This platform is equipped with Exynos 5422 chip, which has four little (A7) cores, four big (A15) cores, a GPU, and other basic components. Built-in sensors

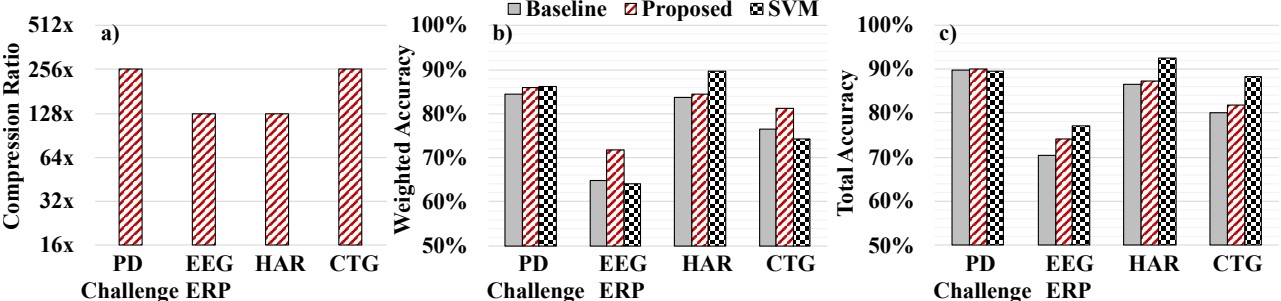

**Figure 4: a) Compression ratio obtained using proposed hypervector design optimization. b) Weighted Accuracy Comparison. c) Total Accuracy Comparison.**

report the power consumption of the big core cluster, little core cluster, GPU and memory separately. We implement both the baseline HDC implementation and the HDC implementation using the level hypervectors obtained by the proposed hypervector design approach using the C programming language. The applications run on a single little core with the lowest frequency setting (200 MHz) since this setup is the closest configuration to a computationally limited edge device. *We note that the model size is independent of this choice, while the relative savings in inference time is comparable to those obtained with other configurations.* We plan to evaluate the proposed technique on low-power embedded processors and custom hardware accelerators as part of our future work.

Table 2 compares the baseline HDC implementation and the proposed approach in terms of the model size, power consumption, and inference time per sample for all applications. For example, for PD Challenge dataset, baseline HDC has a model size of 30 MB using hypervector dimension as $D = 8192$. The proposed approach reduces the number of dimensions to $D = 32$ (i.e., by 256× as shown in Figure 4(a)) and thus, reduces the model size to 120.9 kB (i.e., by 248×). This reduction in model size, which is independent of the hardware, leads to 1.66× reduction in power consumption and 956× reduction in inference time. Since the cores inside the hardware platform are general-purpose cores, we cannot observe a significant decrease in power consumption. The reason behind that is that the core is highly utilized with the lowest frequency setting. However, we can deduce that overall computation and the communication inside the hardware are decreased which yields in a huge boost in inference time. Similarly, the proposed approach reduces the model size by 248×, 126×, 126×, power consumption by 1.79×, 1.68×, 1.58×, and inference time by 742×, 512×, 388× for CTG, HAR, and EEG ERP datasets, respectively. Overall, it achieves more than 600× energy efficiency while reducing the model size from thousands of kB to below or around 100 kB. By evaluating the proposed approach on a commercial hardware, we observe that

### Table 2: Inference evaluation on Odroid XU3. The numbers in parenthesis shows the reduction we obtain using the proposed approach. *PD: PD Challenge, *HDC: Baseline HDC

| | Model Size (kB) | | Power Cons. (mW) | | Inf. Time (ms) | |
|---|---|---|---|---|---|---|
| | HDC* | Proposed | HDC* | Proposed | HDC* | Proposed |
| PD* | 30020 | 120.9 (248×) | 108 | 65 (1.66×) | 239 | 0.25 (956×) |
| CTG | 11208 | 45.1 (248×) | 109 | 61 (1.79×) | 89 | 0.12 (742×) |
| HAR | 17828 | 141.4 (126×) | 109 | 65 (1.68×) | 133 | 0.26 (512×) |
| EEG ERP | 2229 | 17.6 (126×) | 98 | 62 (1.58×) | 155 | 0.40 (388×) |

it can boost the effectiveness of HDC by enabling more than two orders of magnitude reduction in model compression. Hence, we conclude that hypervector design optimization is vital to enable light-weight and accurate HDC on edge devices with stringent energy and computational power constraints.

## 6 CONCLUSION

IoT applications require light-weight learning algorithms to achieve high inference speed and accuracy within the computational power and energy constraints of edge devices. HDC is a computationally efficient learning algorithm due to its simple operations on high dimensional vectors. This paper presented an optimized hypervector design approach to achieve higher accuracy and robustness using a significantly smaller model size. It formulates the hypervector design as a multi-objective optimization problem and presents an efficient algorithm for the first time in literature. Evaluations on four representative health applications show that the proposed approach boosts HDC's effectiveness by achieving more than two orders of magnitude reduction in model size, inference time, and energy consumption while maintaining or increasing baseline HDC accuracy. It also optimizes the trade-off between accuracy and robustness of the model and achieves over 2× higher robustness.

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

# APPENDIX

In Appendix, we provide the mathematical illustration of a simple binary classification example to show the significance of the level dependency.

Let us assume, there are 10 samples $x_s$ evenly distributed in the feature space $F = \{f_1\}$ where $f_1 \in [0, 10]$ as shown in Figure A.1(a). These classes are separated from $f_1 = 5$, such that the class labels $y_s$ are as follows:

$$y_s = \begin{cases} 1 \text{ if } f_1 \leq 5 \\ 2 \text{ otherwise} \end{cases} \quad (A.1)$$

In this problem, $f_1^{min} = 0$ and $f_1^{max} = 10$. We select the first 9 samples for HDC training, and leave the last sample as a query point. We choose quantization level as $M = 10$ such that the quantization intervals from $f_1^1$ to $f_1^{10}$ are $[0, 1), [1, 2), ..., [9, 10]$.

**Figure A.1: Simple binary classification example. a) Initial problem structure. b) Classification using orthogonal hypervectors. c) Classification using hypervectors with level dependency**

**Classification using orthogonal hypervectors as level hypervectors:** We assign random $D$-dimensional bipolar hypervectors $L_1^1$ to $L_1^{10}$ to these intervals. In high dimensions, random hypervectors are nearly orthogonal to each other [10]. Given $x_s$, corresponding sample hypervector becomes $X_s = L_1^{q(x_s^1)}$ where $q(x_s^1)$ is the quantization level that contains value for the feature $f_1$ of the current sample $x_s$. Then, we generate class encoders $E_1$ and $E_2$ as:

$$E_1 = \sum_{s=1}^{5} X_s = \sum_{s=1}^{5} L_1^{q(x_s^1)} \text{ and } E_2 = \sum_{s=6}^{9} X_s = \sum_{s=6}^{9} L_1^{q(x_s^1)} \quad (A.2)$$

Given a query point $Q$ with value 10, corresponding query hypervector becomes $Q = L_1^{q(x_{10}^1)}$. Classification is performed by calculating the cosine similarity between $Q$ and $E_1, E_2$. Since the generated level hypervectors are orthogonal to each other, the dot product operation in the cosine similarity calculation yields a value close to 0, and thus, the decision becomes random and may give wrong results as shown in Figure A.1(b):

$$E_2 \cdot Q = E_1 \cdot Q \approx 0 \quad (A.3)$$

$$\sum_{s=6}^{9} L_1^{q(x_s^1)} \cdot L_1^{q(x_{10}^1)} \approx \sum_{s=1}^{5} L_1^{q(x_s^1)} \cdot L_1^{q(x_{10}^1)} \approx 0 \quad (A.4)$$

**Classification using hypervectors with level dependency as level hypervectors:** A random $D$-dimensional bipolar hypervector $L_1^1$ is assigned to the interval $[0, 1)$. Unlike the previous approach, we follow the baseline HDC implementation explained in Section 3 to obtain the rest of the level hypervectors. Thus, the level dependency between level hypervectors for $f_1$ is preserved. Then, we generate sample hypervectors and calculate the class encoders as in Equation A.2. Finally, classification is performed by calculating the cosine similarity between query hypervector and class encoders. Since the query hypervector is more similar to the sample hypervectors that is used to generate $E_2$ due to preserved level dependency, the classification is correct and the class label of the query point is predicted as 2 as shown in Figure A.1(c):

$$E_2 \cdot Q = \sum_{s=6}^{9} L_1^{q(x_s^1)} \cdot L_1^{q(x_{10}^1)} > E_1 \cdot Q = \sum_{s=1}^{5} L_1^{q(x_s^1)} \cdot L_1^{q(x_{10}^1)} \quad (A.5)$$

