# OpenReview forum: "Hypervector Design for Efficient Hyperdimensional Computing on Edge Devices"
_tinyml.org/tinyML/2021/Research_Symposium — tinyML 2021 Regular_

### Official Review · AnonReviewer2 · 2021-01-30

**Overall Merit Score:** 2

**Brief Summary:**

Hyperdimensional computing (HDC) has arisen as a modern lightweight learning algorithm with lower processing and energy requirements compared to traditional techniques. This paper proposes a strategy for minimizing the hypervector dimension while preserving accuracy and enhancing the robustness of the classifier. They mentioned that they are formulating hypervector architecture as a multi-objective optimization problem which is a novel approach. Their experiments demonstrate that the proposed solution produces a decrease in model scale, inference time and energy consumption by more than two orders of magnitude on a commercial hardware platform.



**Detailed Comments:**

The authors represent the ideas well and present some convincing argument in support of their claims. The geometric approach to map data points in hyperspace is particularly interesting. Along with this, the use of pareto front solutions to obtain a more robust choice of hypervectors can be expanded further into future works and will open up new avenues of research.

However, the work doesn't provide sufficient support for a number of its presentations.
•	The authors present only GPU results as part of the contribution for Edge device implementation. Also, there is scant elaboration for ML/DL models which falls short of the core requirements for TinyML research directives.
•	The authors present inference evaluations of their approach for all four applications in table 2. The results are encouraging in the sense that there is significant reduction in dimension which will translate to model size reduction. However, it is difficult to infer how this approach will fare when compared to deep learning models running on the same platform. This is one aspect where needs to be some more exploration.
•	Also, the authors claim that the reduction in memory size will allow this algorithm to be deployed on to edge devices. Even though they demonstrate GPU performance results on the Odroid XU3 platform, it is not sufficient enough to make a generalization. There should be comparison of the implementation for ARM Cortex and Nvidia GPUs to create a benchmark.


**Paper Strengths:**

•	The flow of the paper is consistent and is well written in terms of specifying objectives.
•	It incorporates novel geometric principles and diagrams to help understand the effect of HDC mapping on hyperspace from the input features.
•	Use of pareto front solutions is particularly interesting as it results in a more robust choice of hypervectors than baseline techniques.
•	By refining the trade-off between accuracy and noise robustness, it achieves 2× higher robustness while retaining the same accuracy.


**Paper Weaknesses:**

•	Contribution in TinyML domain is not clear.
•	Comparison with deep learning models rather than machine learning should be presented.
•	The hardware scope is limited. It needs deployment of the model on different GPU platforms.



**Poster (If Paper Is Rejected):**

1: Yes, ok for poster sesion to nurture work

**Reviewer Confidence:**

5: The reviewer is absolutely certain that the evaluation is correct and very familiar with the relevant literature

---

### Official Review · AnonReviewer3 · 2021-01-30

**Overall Merit Score:** 3

**Brief Summary:**

The paper is related to Hyperdimensional computing. It proposes a technique to decrease the hypervector dimension to redcue the model size and overall HW power cost.

**Detailed Comments:**

Details provided in Strengths and weaknesses sections

**Paper Strengths:**

The paper is very well written and the analysis is clear. The problem statement of reducing HDC model size is well addressed in the paper. The authors present clearly the multi-objective optimization problem and its results.
- Background on HDC is useful and conference attendees will probably appreciate even if this is a very specific application
- HDC is quite promising for lightweight learning algorithm and should be addressed at TinyML
- results are quite convincing by showing tradeoffs between accuracy and robustness
- Even if power consumption reduction is not effectively demonstrated with the HW used (general purpose), the benchmark is interesting and makes the paper even more convincing to me.


**Paper Weaknesses:**

- HDC may be seen as a very specific example of learning algorithms
- table 5 is quite complex to understand. The authors should probably try to explain it correctly if the paper is accepted.
- other similar techniques have been presented before but the paper is very complete showing optimized vector design and benchmarking

**Poster (If Paper Is Rejected):**

1: Yes, ok for poster sesion to nurture work

**Reviewer Confidence:**

3: The reviewer is fairly confident that the evaluation is correct

---

### Official Review · AnonReviewer1 · 2021-01-30

**Overall Merit Score:** 3

**Brief Summary:**

This paper presents a method for optimizing the previously random projections used in hyperdimensional computing (HDC).  By optimizing these projections the size of the models can be reduced by two orders of magnitude.  There appears to be no loss of accuracy.  The models are run on an Odriod processor with 4x A7 cores, 4x A15 cores and a GPU, and performance data is presented.

**Detailed Comments:**

This is a difficult paper to assess because within the limited context which it presents, it is novel, apparently successful, and convincing.  However, it ignores an enormous body of work in cognate and older fields (the most obvious of which is the ELM literature, although that itself has been accused of being not novel by earlier researchers...). I suggest a lukewarm accept, in that the method is valid and well-presented.  The hardware platform is not really what I think of as TinyMl, having eight cores and a GPU, but it is at least real hardware and not a numerical exercise.

**Paper Strengths:**

The optimization method is clearly successful although the underlying algorithm is intrinsically bloated, so it would be surprising if it wasn't successful.  The degree of testing is good, with performance figures for a real processor.

**Paper Weaknesses:**

The real weakness of this paper is that it presents HDC as a unique method (which it is well known not to be) and as such completely ignores a very significant body of work in optimizing similar method such as ELM, No-Prop, and other methods which use random projections.  As such, there is no comparison with the methods of (amongst many others) Thakur and colleagues at IIS, who have developed both algorithms and silicon hardware for optimizing this problem.  If one takes this paper at face value (without knowing about the parallel work) it is novel, but it is a fine example of the approach that everything is novel if you avoid reading the literature.  The data sets addressed are not particularly convincing and I was not convinced that there was not overtraining happening (HDC is somewhat more immune to overfitting than many ML methods, but this optimization method re-introduces that possibility quite strongly).

**Poster (If Paper Is Rejected):**

1: Yes, ok for poster sesion to nurture work

**Reviewer Confidence:**

5: The reviewer is absolutely certain that the evaluation is correct and very familiar with the relevant literature

---

### Official Review · AnonReviewer4 · 2021-01-30

**Overall Merit Score:** 3

**Brief Summary:**

This work aims at reducing the hypervector size in hyperdimensional computing (HDC) while maintaining the accuracy. The hypervector design is formulated as a multi-objective optimization problem. For datasets pertaining to wearable health applications, the hypervector dimension is reduced by >128X, and the proposed algorithm has also been evaluated on a commercial hardware platform.

**Detailed Comments:**

Please clearly define what the “weighted accuracy” is.

The authors only compared to linear SVM, which does not provide high accuracy for ML tasks. Can the authors add a well-trained lightweight DNN as a comparison point in Fig. 4?

Regarding Table 2, does the reported HDC and proposed datapoints have the same accuracy for the 4 datasets? If not, the authors should also report the amount of accuracy degradation for the proposed schemes.

The authors quoted [13][20] for the baseline hypervector size of 8192, but for the wearable health datasets, it is not clear whether hypervector size of 8192 is needed. It is not clear whether the claimed 128X reduction became a large value because the authors choose a sub-optimal baseline. To verify this, the authors can perform simple sweep of the hypervector sizes for the target wearable health datasets, and choose the hypervector size that maintains high accuracy. Then, 4 different datasets will have their own baseline hypervector size, and the vector size obtained by the proposed scheme should be compared against those corresponding size values, different for each dataset.


**Paper Strengths:**

Previously proposed hyperdimensional computing works have used very large vector sizes (in the order of thousands), and this paper proposes systematic methods to reduce such vector sizes.
The proposed scheme has been evaluated on four different distinct datasets targeting wearable health applications, and also validated on the Odroid hardware platform. The reported Pareto front sets of hypervector design provide a trade-off between robustness to noise and overall accuracy.


**Paper Weaknesses:**

The decision of using the hypervector size of 8192 for the baseline has not validated well for the target wearable health datasets in this paper (more comments in the Detailed Comments section below).

**Poster (If Paper Is Rejected):**

1: Yes, ok for poster sesion to nurture work

**Reviewer Confidence:**

4: The reviewer is confident but not absolutely certain that the evaluation is correct

---

### Decision · Program_Chairs · 2021-02-05

**Decision:**

Accept (Regular)

**Comment:**

Congratulations on your paper's acceptance!

Your paper has been accepted as a full-length regular paper.

Please read the reviews carefully and make sure the concerns are addressed in your final submission.

All accepted papers will be given a slot in the TinyML Summit schedule for an oral presentation on Friday, March 26, 2021.

Camera ready instructions will follow soon. All papers will be hosted on arXiv and published papers will have the following header stamp: “Published as a conference paper at TinyML Research Symposium 2021.” The paper will also be presented on the program website.